# MAKEUPANYONE: SELF-SUPERVISED IDENTITY-PRESERVING MAKEUP TRANSFER WITH REGION-AWARE MULTI-SCALE ALIGNMENT

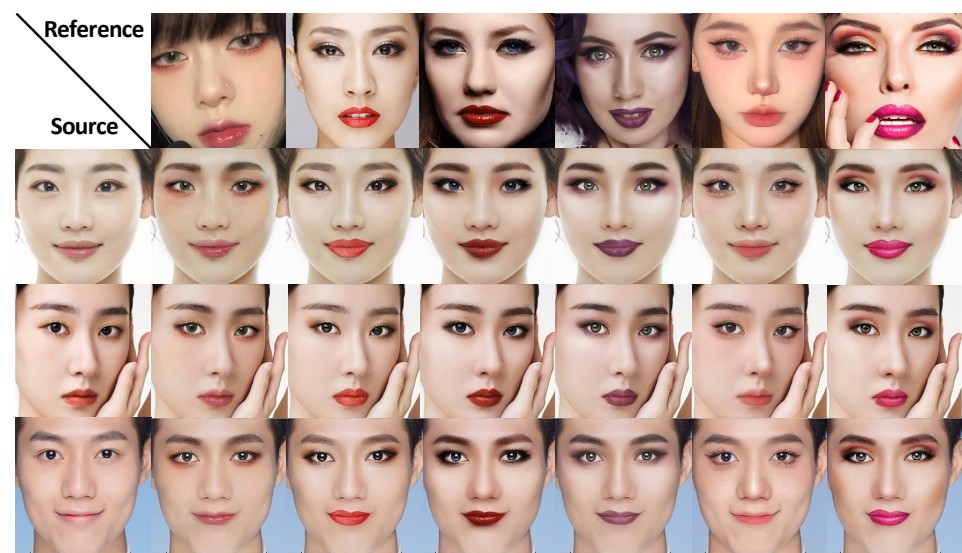

Figure 1: MakeupAnyone is an innovative diffusion-based makeup transfer framework that robustly handles a wide range of real-world makeup styles, ensuring high-quality makeup transfer while effectively preserving facial structure consistency.

## ABSTRACT

Existing makeup transfer methods often fail in real-world scenarios, as the scarcity of high-quality paired data leads to model overfitting and unstable style reproduction, while their poor decoupling of identity from style results in facial distortion and poor identity consistency. To address these challenges, we propose MakeupAnyone, a method that achieves fine-grained, high-fidelity makeup transfer through self-supervised data augmentation and region-aware multiscale alignment. To overcome the lack of paired data, we introduce a self-supervised pipeline that leverages the powerful priors of large Vision Language Models (VLMs) and instruction-guided image editing models for data augmentation and then conducts data filtering based on facial structure consistency, aesthetic quality, and image-text consistency to produce pseudo-makeup pairs with high quality and diversity. Furthermore, we propose a Region-Aware Multi-Scale Alignment approach for makeup feature extraction and training. Specifically, we utilize two distinct Makeup Encoders to respectively capture multi-scale global semantic features and local regional style features. These features are then intelligently fused via an adaptive fusion module. The training is guided by a composite loss function that explicitly balances global style fidelity, local detail accuracy, and identity consistency across facial components Extensive experiments on Makeup Transfer and Makeup-Wild datasets and our newly curated dataset demonstrate that MakeupAnyone achieves state-of-the-art performance with improved detail fidelity and identity similarity.

# 1 INTRODUCTION

Makeup transfer, an essential task in computer vision, aims to realistically and seamlessly apply the makeup style from an arbitrary reference image to a target face while strictly preserving its original identity. This technology holds significant promise for a wide range of applications, including digital entertainment, portrait enhancement, and social media. However, achieving high-fidelity and flexible makeup transfer remains a formidable challenge. The core difficulty lies in the need for a model to precisely disentangle and recombine the highly entangled visual attributes of identity and makeup style, amidst complex variations in facial geometry, lighting conditions, and expressions.

Early approaches, (Jiang et al., 2020; Li et al., 2018; Liu et al., 2021; Deng et al., 2021) predominantly based on Generative Adversarial Networks (GANs), have achieved a degree of success in controlled scenarios.

However, these methods exhibit several inherent limitations when confronted with the complex and diverse makeup styles found in the real world. The first critical issue is identity leakage. GAN-based generators often fail to fully disentangle makeup textures from identity-defining features such as facial geometry and skin tone. This entanglement frequently results in unnatural distortions or altered facial structures in the transferred image, creating a "face-swapping" illusion that severely compromises realism. The second limitation is insufficient style fidelity. When dealing with fine details like sharp eyeliner, glitter, or gradient lip colors, GAN-based models are prone to generating blurry, artifact-ridden, or color-inaccurate results, failing to reproduce the artistry and sophistication of modern cosmetics. Finally, and most critically, existing methods suffer from a heavy reliance on high-quality paired datasets—collections of images featuring the same individual with and without makeup under identical lighting and pose. The acquisition of such datasets is prohibitively expensive and inherently limited in scale, which restricts the model's generalization capabilities and hinders its performance on diverse, in-the-wild inputs.

In recent years, Diffusion Models (Rombach et al., 2022; Podell et al., 2023; Labs et al., 2025) have emerged as a powerful alternative, garnering significant attention for their exceptional fidelity and diversity in image generation tasks and offering new possibilities for overcoming these challenges. Despite their potential, directly applying diffusion models to makeup transfer is non-trivial. Key challenges remain, particularly in achieving fine-grained control over local regions like the eyes and lips, and in fundamentally ensuring a complete decoupling of identity and style.

To address these multifaceted challenges, we introduce MakeupAnyone, a novel self-supervised, identity-preserving makeup transfer framework. Our framework tackles the data bottleneck by introducing a self-supervised pipeline that leverages the generative power of large-scale vision-language models to create a vast, high-quality pseudo-paired dataset, obviating the need for real paired data. Concurrently, it incorporates a novel Region-Aware Multi-Scale Alignment architecture that adeptly captures both global semantics and intricate local details, enabling meticulous control over makeup application while ensuring high-fidelity identity preservation. Our work aims to advance makeup transfer technology to a new level of practicality and robustness.

The main contributions of this work can be summarized as follows:

- We propose a self-supervised data augmentation pipeline to address the scarcity of high-quality paired data. By leveraging large Vision-Language Models (VLMs) and instruction-guided image editing models, our pipeline automatically generates diverse pseudo-makeup pairs, which are subsequently filtered for facial structure consistency, aesthetic quality, and image-text consistency to build a robust training dataset.

- We design a novel Region-Aware Multi-Scale Alignment architecture. It employs parallel Makeup Semantic Encoder and Region Style Encoder to capture global makeup semantics and region-aware style features, respectively. These features are then integrated by an adaptive fusion module, enabling a more precise decoupling of identity from makeup style for fine-grained transfer.

- We formulate a composite training loss function to ensure high-fidelity results. This function explicitly balances global style fidelity, local detail accuracy, and identity consistency, guiding the model to accurately reproduce makeup styles while preserving the original facial structure and effectively preventing identity distortion.

- Extensive experiments on public datasets and our newly curated dataset demonstrate that our method outperforms previous state-of-the-art approaches in terms of both makeup detail fidelity and identity similarity.

## 2 RELATED WORK

### 2.1 MAKEUP TRANSFER

Over the past decade, makeup transfer (Tong et al., 2007; Guo & Sim, 2009) has continued to attract attention. Early methods were mostly GAN-based (Goodfellow et al., 2014). For example, BeautyGAN (Li et al., 2018) used a dual-input-output generator for makeup application and removal. PairedCycleGAN (Chang et al., 2018) designed a style discriminator for local consistency, and BeautyGlow (Chen et al., 2019) decoupled makeup and identity latent variables based on the Glow (Kingma & Dhariwal, 2018) framework. To solve misalignment, PSGAN (Jiang et al., 2020; Liu et al., 2021) introduced an attention mechanism for feature alignment, while SCGAN (Deng et al., 2021) encoded makeup features into spatially invariant style vectors. Later works focused on specific improvements. RamGAN (Xiang et al., 2022) and SpMT (Zhu et al., 2022) used local attention to alleviate interference between components. Others like FAT (Wan et al., 2022), SSAT (Sun et al., 2022; 2023), and EleGANt (Yang et al., 2022) improved pseudo-paired data synthesis using geometric transformation and fusion strategies. For complex styles, LAND (Gu et al., 2019) used local discriminators for details, while CPM (Nguyen et al., 2021b) leveraged semantic mapping for structural alignment. The performance of these methods heavily relies on the quality of pseudo-paired data used for training. Therefore, improving the generation strategy has been a key focus (Chang et al., 2018; Sun et al., 2022; Yang et al., 2022; Wan et al., 2022). Recently, Stable-Makeup (Zhang et al., 2024) leveraged Stable Diffusion (Rombach et al., 2022) and GPT-4V to improve the realism and consistency of pseudo-paired data, advancing the state of the art. MakeupAnyone advances this direction through a self-supervised data augmentation pipeline. We automatically synthesize and filter high-quality pseudo pairs, which reduces dependence on manual curation and provides a robust data foundation for our model.

### 2.2 DIFFUSION-BASED IMAGE GENERATION

Diffusion models have excelled in multimodal image generation, with wide applications in tasks such as text-to-image (Podell et al., 2023; Ramesh et al., 2022; Rombach et al., 2022; Saharia et al., 2022), image editing (Li et al., 2023; Mou et al., 2023a; Tsaban & Passos, 2023; Xie et al., 2023; Zhang et al., 2023b), and controllable generation (Ma et al., 2023; Mou et al., 2023b; Zhang & Agrawala, 2023; Zhao et al., 2023). Originating from the physical inverse diffusion process (Sohl-Dickstein et al., 2015), these models recover a clear image from noise by gradual denoising. Seminal works like DDPM (Ho et al., 2020) verified their generation feasibility, while DDIM (Song et al., 2020a) improved sampling efficiency through non-Markov inference. To reduce computational cost, Latent Diffusion Models (LDMs) (Rombach et al., 2022) use an autoencoder to perform diffusion in a compressed latent space. This strategy, central to powerful models like Imagen (Saharia et al., 2022) and Stable Diffusion (Zhang et al., 2024), balances efficiency and quality. Stable Diffusion XL (Podell et al., 2023) further optimizes detail performance and color consistency through a two-level structure. For generation control, ControlNet (Zhang et al., 2023a) enables fine-grained structural constraints using conditions like edges, depth, and human posture. For customization, Dream-Booth (Ruiz et al., 2023) improves concept consistency through subject-specific tuning at a high training cost. In image editing, methods like Ledits (Tsaban & Passos, 2023) achieve zero-shot editing, adding new flexibility. The diffusion-based paradigm is now mainstream in image generation, achieving leading performance due to its high-quality modeling and generalization. Recognizing this potential, our work introduces a diffusion model to the makeup transfer task, constructing an efficient and robust framework. Our method demonstrates excellent performance on complex makeup styles, showing stronger expressiveness and style fidelity than traditional GAN-based methods.

## 3 METHODS

In this section, we present the details of MakeupAnyone, our proposed framework for self-supervised, identity-preserving makeup transfer. Our method is built upon a conditional diffusion model that takes a no-makeup image as input and progressively generates a high-quality makeup effect through a Denoising U-Net, guided by precise makeup style features. To systematically address the challenges of existing approaches, we have designed a complete pipeline composed of three core components, with the overall framework illustrated in Figure 2.

First, we overcome the dependency on paired data by introducing a Self-supervised Data Augmentation Pipeline (depicted in Section 3.1) to automatically generate high-quality training samples. Second, we propose a novel Region-Aware Multi-Scale Alignment Architecture (detailed in **??**) to

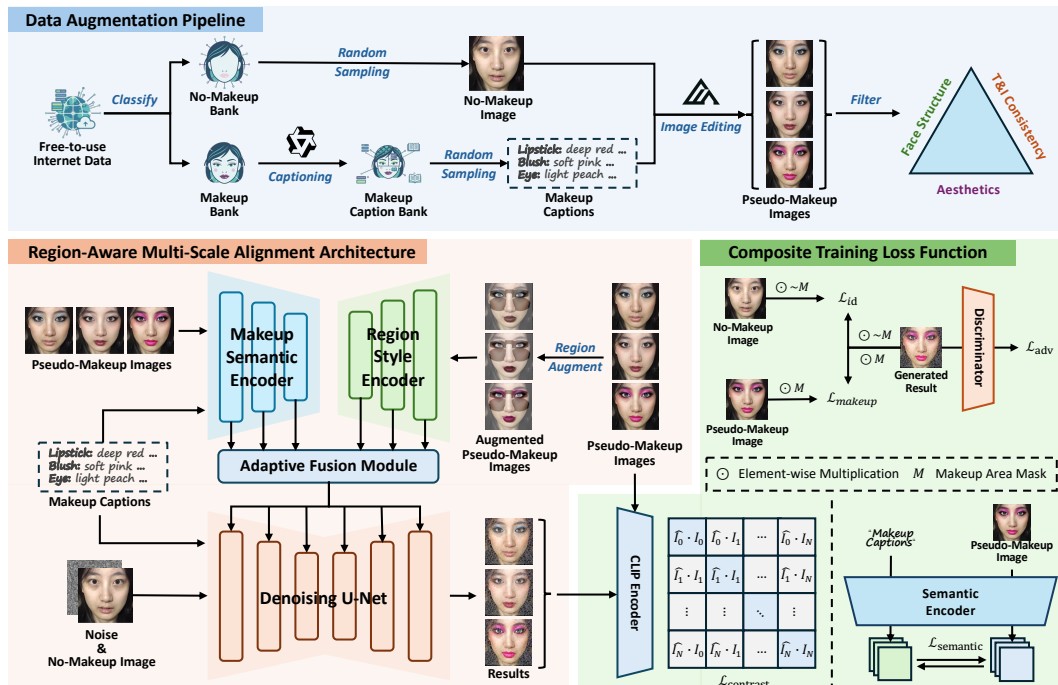

Figure 2: Overview of MakeupAnyone. Our framework achieves photorealistic, identity-preserving, and text-controlled virtual try-on through an automated data pipeline, a region-aware dual-encoder architecture, and a composite loss function.

precisely decouple and control the makeup style across its global semantics and local details. Finally, we formulate a Composite Training Loss Function (as illustrated in Section 3.3) to collaboratively optimize identity consistency, style fidelity, and generation realism, thereby ensuring high-quality transfer. These three components are elaborated upon in detail in the following sections.

## 3.1 SELF-SUPERVISED DATA AUGMENTATION PIPELINE

From a data perspective, the datasets relied upon by mainstream makeup transfer research often suffer from limited sample sizes and a lack of stylistic diversity, which severely restricts the model's generalization ability in real-world scenarios.

**Free-to-use Internet Data.** To alleviate this bottleneck, we first constructed a large-scale, high-resolution, unpaired dataset by collecting 77,717 makeup images and 33,933 non-makeup images from public free-to-use platforms. The makeup images cover a wide range of styles from daily light makeup to heavy artistic makeup. Based on this, we designed an automated data augmentation pipeline to generate a large-scale, high-quality dataset of pseudo-paired triplets (source image, text description, target image).

**Makeup and Non-Makeup Classification.** To accurately separate our collected data, we trained a dedicated binary classification model. Based on the efficient yet powerful EfficientNetV2 (Tan & Le, 2021) architecture, the model is fine-tuned to distinguish between images with and without makeup. After training, we applied this classifier to our entire collection of 111,650 images, automatically sorting them into the "Makeup Bank" and "No-Makeup Bank" with high precision, forming the foundation for the subsequent steps.

**Structured Makeup Caption Bank.** To create a bank of fine-grained, disentangled instructions, we leverage the powerful Large Vision Model, Qwen2.5-VL (Bai et al., 2025), to process the images in our "Makeup Bank." The VLM automatically generates structured makeup descriptions where the overall style is decoupled into key facial regions (e.g., lips, eyes, cheeks). For instance, a description might be formatted as {"lipstick": "matte crimson red ...", "eyeshadow": "smoky black with silver glitter ..."}. This process results in a comprehensive "Makeup Caption Bank" that provides precise and locally-aware guidance.

**Pseudo-Pair Synthesis.** Next, we employ the instruction-guided image editing model, FLUX.1-Kontext (Labs et al., 2025), to generate the pseudo-makeup images. For each synthesis step, we randomly sample a source face from our "No-Makeup Bank" and a structured caption from the "Makeup Caption Bank." The model takes both as input—the image as the base and the caption as the editing instruction—and generates a corresponding made-up face. This procedure forms a pseudo-paired triplet (source, text, target), directly linking a non-makeup source to a text-guided makeup target.

**Quality Control and Filtering.** Finally, to ensure the quality and reliability of our synthetic data, every generated triplet undergoes a rigorous, automated filtering process. We evaluate each sample based on three core criteria, and only those that achieve high scores across all dimensions are retained. Specifically, we use: 1) ArcFace (Deng et al., 2022) to ensure high cosine similarity in identity embeddings for face structure consistency; 2) a pre-trained aesthetic scoring model (Schuhmann, 2022) to filter out images with artifacts or low visual appeal; and 3) CLIP (Ramesh et al., 2022) to verify a strong semantic alignment between the generated image and the text instruction.

## 3.2 REGION-AWARE MULTI-SCALE ALIGNMENT ARCHITECTURE

To achieve precise decoupling and transfer of makeup styles, we propose a novel Region-Aware Multi-Scale Alignment Architecture, as illustrated in Figure 2. The core idea is to extract makeup features through two parallel, complementary streams—one for global semantics and one for local details—and then intelligently fuse them to guide the main generation network. The architecture consists of a dual-stream feature extractor, an adaptive fusion module, and a denoising U-Net that generates the final result. To capture the rich characteristics of a given makeup style, we employ two parallel encoders that process the pseudo-makeup reference image from different perspectives.

**Makeup Semantic Encoder.** This encoder is responsible for capturing the semantic style features from the reference image which is aligned with makeup captions. To leverage powerful pre-trained visual priors, we instantiate this Encoder using the encoder part of the Denoising U-Net. It takes both the pseudo-makeup image and its corresponding makeup caption as input, allowing it to extract style features that are grounded in textual semantics, thus producing a more abstract and robust style representation.

**Region Style Encoder.** In parallel, a lightweight Region Style Encoder focuses on extracting fine-grained, local style features focusing on the specific makeup areas. It is composed of a simple stack of convolutional layers. As shown in Figure 2, we apply region-based augmentations with the pre-processed makeup area mask to its input. This encourages the encoder to learn robust representations of regional styles rather than overfitting to specific textures, enhancing its generalization ability.

**Adaptive Fusion Module.** This module is designed to merge the above two feature streams at multiple scales. For each scale, the fusion process is as follows: first, the semantic features and regional style features are concatenated. Then, spatial and channel attention maps are computed from these concatenated features. These attention maps are applied back to both feature streams to highlight the most salient stylistic information in each. Finally, a lightweight convolutional network predicts a dynamic fusion weight $\alpha \in (0, 1)$, and the two attended feature maps are combined in a weighted sum to produce the final fused makeup feature $\mathbf{F}^{\text{fused}}$. This ensures a comprehensive style representation that balances overall harmony with intricate details.

**Cross-Attention based Denoising Generation.** The main Denoising U-Net takes the noise and f the non-makeup source image as inputs. The fused makeup features $\mathbf{F}^{\text{fused}}$ are injected into the corresponding layers of the U-Net to guide the denoising process. This injection is achieved via cross-attention mechanisms. At each corresponding layer, the U-Net's internal feature maps act as the *Query*, while the fused makeup features $\mathbf{F}^{\text{fused}}$ serve as the *Key* and *Value*.

## 3.3 COMPOSITE TRAINING LOSS FUNCTION

To effectively train our network and balance the multiple objectives of identity preservation, style fidelity, and photorealism, we designed a composite loss function. This function is a weighted sum of five distinct loss terms, each targeting a specific aspect of the makeup transfer task. The overall objective is formulated as:

$$\mathcal{L}_{\text{total}} = \lambda_{\text{makeup}}\mathcal{L}_{\text{makeup}} + \lambda_{\text{adv}}\mathcal{L}_{\text{adv}} + \lambda_{\text{id}}\mathcal{L}_{\text{id}} + \lambda_{\text{semantic}}\mathcal{L}_{\text{semantic}} + \lambda_{\text{contrast}}\mathcal{L}_{\text{contrast}}, \tag{1}$$

where the $\lambda$ terms are hyperparameters that balance the contribution of each component.

**Makeup Reconstruction Loss ($\mathcal{L}_{\textbf{makeup}}$).**  This is a pixel-level loss to ensure the accuracy of the transferred makeup. Both the generated result $I^{\text{gen}}$ and the pseudo-makeup target image $I^{\text{tgt}}$ are multiplied by a makeup area mask ($M$), and we compute the L1 distance between them. Its purpose is to directly force the generator to learn the precise color, shape, and placement of the makeup, serving as the primary supervision for style reconstruction. The loss is defined as:

$$\mathcal{L}_{\text{makeup}} = \mathbb{E}\left[\left\|(I^{\text{gen}} \odot M) - (I^{\text{tgt}} \odot M)\right\|_1\right]. \tag{2}$$

**Adversarial Loss ($\mathcal{L}_{\textbf{adv}}$).**  To enhance the photorealism of the generated makeup, we employ an adversarial training scheme with a discriminator $D$. The generator $G$ is trained to produce results that can fool this discriminator. The purpose of this loss is to push the generated images to be perceptually indistinguishable from real ones, preventing blurry artifacts. The objective is formulated as a non-saturating adversarial loss:

$$\mathcal{L}_{\text{adv}} = \mathbb{E}_{I^{\text{src}},\text{cond}}\left[-\log D(G(I^{\text{src}}, \text{cond}))\right]. \tag{3}$$

**Identity Preservation Loss ($\mathcal{L}_{\textbf{id}}$).**  Preserving the subject's identity is crucial. Our framework achieves this by ensuring that the non-makeup regions of the generated image align with those of the original non-makeup source image. As illustrated in Figure 2, we use the inverse of the makeup area mask ($1 - M$) to isolate the areas without makeup. We then compute a L1 loss between these corresponding non-makeup regions. This is formulated as:

$$\mathcal{L}_{\text{id}} = \mathbb{E}\left[\left\|(I^{\text{gen}} \odot (1 - M)) - (I^{\text{src}} \odot (1 - M))\right\|_1\right], \tag{4}$$

where $I^{\text{gen}}$ is the generated result, $I^{\text{src}}$ is the source non-makeup image, and $M$ is the makeup area mask.

**Semantic Loss ($\mathcal{L}_{\textbf{semantic}}$).**  The semantic loss is designed to regularize our Makeup Semantic Encoder ($E_{\text{sem}}$) to ensure that it learns a robust visual representation of the makeup style, where the textual caption acts as a semantic guide rather than the sole source of information. To achieve this, we constrain the encoder's output features to be consistent, whether or not textual conditioning is present. Specifically, for the same target pseudo-makeup image $I^{\text{tgt}}$, we perform two forward passes through the encoder: one with the corresponding makeup caption $T$, yielding the text-conditioned feature $E_{\text{sem}}(I^{\text{tgt}}, T)$, and another with a null or empty text prompt ($\varnothing$), yielding a purely visual feature $E_{\text{sem}}(I^{\text{tgt}}, \varnothing)$. We then minimize the distance between these two feature representations. The purpose of this loss is to force the encoder to primarily rely on the visual information from the image to extract the core makeup style. This prevents the model from "hallucinating" or over-relying on text, making the learned style features more robust and faithful to the reference image. The loss is formulated as the squared L2 distance:

$$\mathcal{L}_{\text{semantic}} = \mathbb{E}\left[\left\|E_{\text{sem}}(I^{\text{tgt}}, T) - E_{\text{sem}}(I^{\text{tgt}}, \varnothing)\right\|_2^2\right]. \tag{5}$$

**Contrastive Loss ($\mathcal{L}_{\textbf{contrast}}$).**  To achieve better disentanglement of style and identity, we incorporate an InfoNCE contrastive loss using a pre-trained CLIP image encoder $E_{\text{clip}}$. In the CLIP embedding space, this loss pulls the generated image (anchor $a$) closer to its positive pair (the target style $p$) while pushing it away from a set of negative samples $\mathcal{N}$. Its purpose is to encourage the model to learn a representation that is sensitive to the specific makeup style. The loss is given by:

$$\mathcal{L}_{\text{contrast}} = -\log \frac{\exp(\text{sim}(a,p)/\tau)}{\exp(\text{sim}(a,p)/\tau) + \sum_{n \in \mathcal{N}} \exp(\text{sim}(a,n)/\tau)}, \tag{6}$$

where $a = E_{\text{clip}}(I^{\text{gen}})$, $p = E_{\text{clip}}(I^{\text{tgt}})$, $n = E_{\text{clip}}(I^{\text{neg}})$, $\text{sim}(\cdot, \cdot)$ is the cosine similarity, and $\tau$ is a temperature hyperparameter.

## 4 EXPERIMENTS

### 4.1 DATASETS

Our experiments are conducted on a combination of existing and newly collected datasets to ensure comprehensive evaluation. We utilize the Makeup Transfer Dataset (Li et al., 2018), which comprises 1115 non-makeup and 2719 makeup images at an approximate $361 \times 361$ resolution, following the established training and testing split of prior work. To assess model robustness against real-world challenges, we also employ the Makeup-Wild Dataset (Jiang et al., 2020), containing 369 non-makeup and 403 makeup images at $256 \times 256$ resolution, which is notable for its significant variations in pose, expression, and background. To further probe generalization and enable cross-domain analysis, we curated a new large-scale, high-quality dataset, herein referred to as Ours Dataset. This collection consists of 33,933 non-makeup and 77,717 makeup images, all at a uniform $512 \times 512$ resolution and featuring rich diversity in pose and expression.

Table 1: Quantitative comparison with state-of-the-art methods on the Makeup Transfer (Li et al., 2018) and Makeup-Wild (Jiang et al., 2020) datasets. For each metric, the **best** result is in bold, and the second-best is underlined. ↑ indicates higher is better, while ↓ indicates lower is better.

| Method | Makeup Transfer | | | | | Makeup-Wild | | | | |
|---|---|---|---|---|---|---|---|---|---|---|
| | SSIM↑ | CLS↑ | L2-M↓ | LPIPS↓ | PSNR↑ | SSIM↑ | CLS↑ | L2-M↓ | LPIPS↓ | PSNR↑ |
| BeautyGAN | 0.870 | 0.830 | 12.270 | 0.490 | 19.750 | 0.870 | 0.850 | 12.030 | 0.550 | 19.910 |
| CPM | 0.674 | 0.568 | 12.200 | 0.504 | 16.638 | 0.634 | 0.507 | 12.377 | 0.560 | 15.965 |
| SPMT | 0.771 | 0.834 | 12.220 | 0.474 | 17.738 | 0.352 | 0.096 | 12.716 | 0.565 | 9.125 |
| SSAT | 0.765 | 0.718 | 14.010 | 0.533 | 17.440 | 0.770 | 0.716 | 13.090 | 0.590 | 17.991 |
| PSGAN | 0.660 | 0.810 | 12.090 | 0.498 | 16.670 | 0.530 | 0.800 | **11.660** | 0.550 | 16.450 |
| EleGANt | 0.650 | 0.830 | 12.130 | 0.490 | 16.600 | 0.520 | 0.810 | 11.750 | 0.540 | 16.180 |
| SCGAN | 0.874 | **0.905** | 12.430 | 0.496 | 19.268 | 0.874 | **0.905** | 11.897 | 0.543 | 20.292 |
| Stable-Makeup | 0.789 | 0.590 | 12.520 | 0.480 | 22.060 | 0.303 | 0.113 | 12.598 | 0.568 | 9.057 |
| SHMT | 0.825 | 0.858 | 12.570 | 0.502 | 20.061 | 0.290 | 0.094 | 13.521 | 0.579 | 9.050 |
| **MakeupAnyone** | **0.895** | 0.874 | **12.010** | **0.414** | **23.705** | **0.899** | **0.905** | 12.240 | **0.531** | **26.466** |

## 4.2 IMPLEMENTATION DETAILS

We build upon the pretrained Instruct-Pix2Pix model (Brooks et al., 2023) and fine-tune our entire framework end-to-end at a resolution of $512 \times 512$ pixels. We use the AdamW optimizer (Loshchilov & Hutter, 2019) with a generator learning rate of $8 \times 10^{-5}$, a weight decay of 0.01, and a cosine annealing schedule. Training is performed on four NVIDIA H20 GPUs with a total batch size of 64 for 100K steps, utilizing gradient clipping at 1.0. Data augmentation includes random horizontal flipping and mild color jitter, and we apply text dropout with a probability of 0.1. For contrastive learning, 8 negatives are sampled within the same identity. At inference time, we employ a DDIM sampler Song et al. (2020b) with 50 sampling steps and a guidance scale of 7.5, following a single inversion pass to anchor the source identity.

## 4.3 EVALUATION METRICS

To quantitatively evaluate our method, we assess three key aspects: identity preservation, style transfer fidelity, and overall perceptual quality. For identity preservation, we compute the Structural Similarity Index Measure (SSIM) Wang et al. (2004), Peak Signal-to-Noise Ratio (PSNR) between the output $\hat{I}$ and source $I_{src}$, and a cosine similarity score (CLS) from the pretrained ArcFace verifier Deng et al. (2022). Higher values for these metrics signify better performance. To evaluate style transfer and perceptual fidelity, we report the Learned Perceptual Image Patch Similarity (LPIPS) Zhang et al. (2018) between the output $\hat{I}$ and reference $I_{ref}$, as well as a proposed masked feature-space $\ell_2$ distance (**L2-M**) (Zhang et al., 2024). L2-M uses a pretrained ResNet-50 He et al. (2016) to compute the mean squared error on embeddings extracted exclusively from makeup regions defined by a face-parsing mask. Lower LPIPS and L2-M scores indicate superior results. Unless otherwise stated, all reported scores are the average of pairwise evaluations over all source–reference combinations in the test set.

## 4.4 QUANTITATIVE COMPARISON

We conduct a comprehensive quantitative comparison against representative makeup transfer baselines, including BeautyGAN (Li et al., 2018), CPM (Nguyen et al., 2021a), PSGAN (Jiang et al., 2020), EleGANt (Yang et al., 2022), SCGAN (Deng et al., 2021), SPMT (Zhu et al., 2022), SSATv (Sun et al., 2022), Stable-Makeup (Zhang et al., 2024), and SHMT (Sun et al., 2024). The results, presented in Table 2 and Table 1, demonstrate that our method outperform all baselines across the three evaluated datasets. On both the

Table 2: Quantitative comparison with state-of-the-art methods on our proposed dataset. For each metric, the **best** result is in bold, and the second-best is underlined. ↑ indicates higher is better, while ↓ indicates lower is better.

| Method | SSIM↑ | CLS↑ | L2-M↓ | LPIPS↓ | PSNR↑ |
|---|---|---|---|---|---|
| BeautyGAN | 0.804 | 0.845 | 11.277 | 0.484 | 20.954 |
| CPM | 0.770 | 0.612 | 11.444 | 0.477 | 17.124 |
| PSGAN | 0.827 | 0.882 | 11.494 | 0.482 | 19.132 |
| EleGANt | 0.339 | 0.0958 | 12.214 | 0.560 | 10.163 |
| SCGAN | 0.826 | 0.858 | 12.035 | 0.501 | 17.887 |
| SPMT | 0.817 | 0.882 | 11.542 | 0.473 | 18.810 |
| SSAT | 0.868 | 0.838 | 12.206 | 0.504 | 21.344 |
| Stable-Makeup | 0.747 | 0.675 | 10.626 | 0.479 | 21.400 |
| SHMT | 0.848 | **0.8828** | 11.467 | 0.489 | 19.825 |
| **MakeupAnyone** | **0.878** | 0.862 | **9.876** | **0.426** | **24.695** |

| Source | Reference | | BeautyGAN | CPM | EleGANt | PSGAN | SCGAN | SHMT | SpMT | SSAT | StableMakeup | Ours |

Figure 3: Qualitative comparison with makeup transfer methods. The results show that our model generates more realistic images with finer details and fewer artifacts.

MT dataset and the more challenging Makeup-Wild dataset, our approach shows a clear superiority. This indicates that our model achieves a better balance of preserving the identity and facial structure while delivering higher fidelity and accuracy in style transfer. This advantage is particularly pronounced on the Makeup-Wild and our proposed datasets, where our method's robust performance highlights its ability to handle significant real-world variations in pose, lighting, and expression—a common failure point for previous methods.

We also compared the makeup transfer results with several other methods, including Doubao, Gemini, GPT5, and FLUX. Compared to these methods, our approach is able to maintain facial structure consistency while accurately reproducing makeup details, especially when transferring complex and extreme makeup styles. Methods such as Doubao, Gemini, and GPT5 failed to effectively preserve facial structure, leading to noticeable changes in facial features. Although FLUX can maintain facial structure, its makeup details and color accuracy are suboptimal. Overall, our method demonstrates stronger robustness in terms of detail fidelity and identity consistency, and it is better equipped to handle diverse and extreme makeup styles.

## 4.5 QUALITATIVE COMPARISON

We further conducted qualitative experiments to analyze the performance of our method across various makeup styles. Figure 3 and Figure 4 show a comparison with existing methods and close-source image editing models, highlighting the advantages of our method in terms of detail fidelity and identity consistency. Compared to traditional generative adversarial network (GAN) methods, MakeupAnyone accurately reproduces complex makeup details, avoiding the common "face-swapping" phenomenon. Compared to other diffusion-based methods, our method not only maintains high makeup transfer quality but

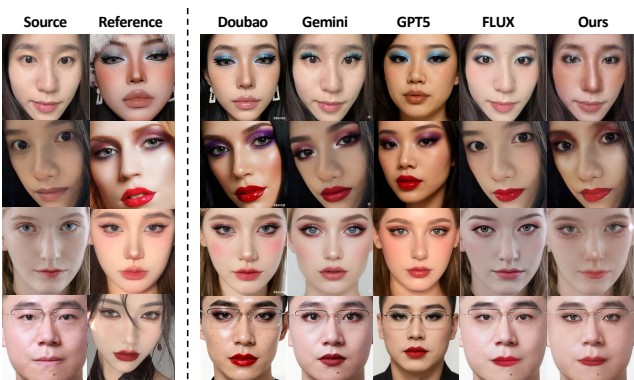

Figure 4: Virtual makeup try-on comparison with close-source image editing models. Our model is compared against leading models such as Doubao, Gemini, GPT5, and FLUX.

also better preserves original facial features. Furthermore, MakeupAnyone demonstrates greater robustness when handling extreme makeup styles, particularly complex styles such as detailed eye makeup and gradient lipstick shades.

Figure 5: Qualitative ablation study results. The figure shows the impact of removing different modules on makeup transfer performance. Each column presents the progressive ablation results by removing modules such as the makeup semantic encoder, region encoder, contrastive loss, adversarial loss, makeup loss, identity loss, and semantic loss.

## 4.6 ABLATION STUDIES

We conducted extensive ablation studies to validate each component, with quantitative and qualitative results shown in Table 3 and Figure 5, respectively. The results confirm that every module is essential. Specifically, removing either the Makeup Semantic Encoder or the Region Style Encoder impairs the model's ability to capture global style and fine-grained local details. Similarly, ablating the loss functions demonstrates their distinct roles: the makeup and identity losses are vital for accurate style transfer and preventing facial distortion; the adversarial and contrastive losses enhance perceptual realism; and the semantic loss is crucial for maintaining color consistency.

Table 3: Ablation study of our proposed method. The performance of the full model is shown in the last row for comparison.

| Method | SSIM↑ | CLS↑ | L2-M↓ | LPIPS↓ | PSNR↑ |
|---|---|---|---|---|---|
| w/o Makeup Semantic Encoder | 0.699 | 0.746 | 12.318 | 0.551 | 10.862 |
| w/o Region Style Encoder | 0.873 | 0.829 | 10.924 | 0.473 | 23.792 |
| w/o Contrastive Loss | 0.858 | 0.814 | 11.072 | 0.480 | 23.181 |
| w/o Adversarial Loss | 0.854 | 0.809 | 11.104 | 0.475 | 23.815 |
| w/o Makeup Loss | 0.842 | 0.831 | 11.889 | 0.498 | 23.710 |
| w/o Identity Loss | 0.873 | 0.823 | 12.125 | 0.504 | 23.103 |
| w/o Semantic Loss | 0.874 | 0.850 | 12.302 | 0.505 | 24.263 |
| **MakeupAnyone (Full Model)** | **0.878** | **0.862** | **9.876** | **0.426** | **24.695** |

## 5 CONCLUSION

In this paper, we introduced MakeupAnyone, a novel framework designed to address the critical challenges of identity distortion, insufficient style fidelity, and data scarcity that hinder existing makeup transfer methods. Our solution is threefold. First, we tackle the data bottleneck with a self-supervised pipeline that leverages large generative models to create a vast and diverse pseudo-paired dataset, eliminating the need for expensive real-world data collection. Second, our proposed Region-Aware Multi-Scale Alignment architecture, featuring parallel semantic and style encoders, achieves a more precise decoupling of identity and makeup style for fine-grained control. Finally, a carefully designed composite loss function ensures that the model is optimized for multiple objectives simultaneously, from photorealism and local detail accuracy to global identity preservation. Extensive experiments on multiple datasets demonstrate that MakeupAnyone significantly outperforms previous state-of-the-art methods, yielding results with superior makeup detail fidelity, stronger identity consistency, and greater robustness on in-the-wild images. By effectively combining self-supervised data generation with a sophisticated feature alignment architecture, MakeupAnyone not only sets a new benchmark for makeup transfer but also offers a promising paradigm for other fine-grained image style transfer tasks.

**Limitations and Future Work.** Despite its state-of-the-art performance, our method has limitations, such as the high computational cost of the data generation pipeline and its reliance on the performance of pre-trained models. Future work will focus on improving computational efficiency to support real-time video applications. Additionally, enhancing user controllability, such as allowing interactive adjustments of makeup intensity, is a valuable direction for future research.

## ETHICS STATEMENT

We recognize the important ethical considerations involved in this research. Our dataset is derived from public images; while we only use and release AI-generated "pseudo-makeup" images to protect individual privacy, the source data and the pre-trained models we leverage may contain demographic biases, potentially leading to varied performance across different populations. We acknowledge the risk that this technology could be misused for creating misleading content (deepfakes) and may reinforce narrow societal beauty standards. We strongly condemn any malicious use and consider addressing these ethical challenges a key part of our future work.

## REPRODUCIBILITY STATEMENT

To ensure reproducibility, we provide this anonymous link, which includes our source code and model weights. The methods described in detail in Section 3 offer a clear technical blueprint for implementing our architecture. Specific implementation details, such as training configurations, optimizer settings, and hyperparameters, are provided in Section 4.2. We believe that using the information in these sections and the provided code, an independent researcher can reproduce our experimental results presented in Section 4.4 and Section 4.6.

## LLM USAGE

We acknowledge the use of LLMs as a writing assistant. The LLMs are utilized solely to assist with proofreading, grammatical correction, and minor stylistic refinements of the manuscript.

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

## A    APPENDIX

