# OpenReview forum: "MakeupAnyone: Self-Supervised Identity-Preserving MakeUp Transfer with Region-Aware Multi-Scale Alignment"
_ICLR.cc/2026/Conference — Submitted to ICLR 2026_

### Official Review · Reviewer_MY1A · 2025-10-16

**Soundness:** 1
**Presentation:** 2
**Contribution:** 2
**Rating:** 2
**Confidence:** 5

**Summary:**

The paper MakeupAnyone proposes a diffusion-based framework for identity-preserving makeup transfer. The method uses a dual-encoder structure to capture global and local style information and applies a composite loss to balance realism and identity preservation. The authors also introduce an automatic data preparation pipeline and claim that it enables self-supervised learning. Experiments are conducted on several existing datasets and one newly collected dataset, showing numerical improvements over previous methods.

**Strengths:**

・The paper provides a reasonably clear system overview and includes architectural details.

・Experimental results are reported across datasets with standard metrics and ablation studies.

・Visual results indicate that the framework can perform basic makeup transfer with acceptable quality under standard conditions.

**Weaknesses:**

・The authors state that they aim to solve the problem of lacking paired data by adopting a self-supervised approach. However, if my understanding is correct, the authors do not provide an algorithmically self-supervised solution. Instead, they still generate paired data first and then train the model in a supervised manner using these pairs. About the dataset, the only difference from Stable-Makeup lies in the fact that the latter manually selected its data, whereas more recent studies such as BeautyBank have provided synthetic datasets, and FLUX-Makeup and EvoMakeup have also proposed large-scale data generation and automatic filtering pipelines. In addition, FFHQ-Makeup has released an automatically generated paired makeup dataset. Although the latter three works are still in preprint status, the authors’ claim that their method is self-supervised is not reasonable.

・The sources of the collected data are not clearly indicated regarding whether authorization was obtained, whether the dataset will be made public, or how the dataset was constructed and evaluated. Moreover, I noticed that several images also appear in BeautyREC. Although it is possible that the authors obtained them from the same sources, the lack of citation to that work is inappropriate.

・The authors’ region-based augmentations rely on the use of masks. This design might limit performance — for example, are eyeshadow and blush always guaranteed to exist entirely within the mask? If the mask is inaccurate, could it truncate gradient colors? This raises concerns that the method might struggle to handle dramatic makeup transfer cases, whereas Stable-Makeup can manage them more effectively.

・During the experimental phase, the authors only used datasets from BeautyGAN and PSGAN, while omitting others such as BeautyFace and LADN, which contain a broader variety of makeup styles and heavy-makeup images. For the dataset collected by the authors themselves, they did not provide any statistical analysis or evaluation, making it impossible to assess the dataset’s quality. This lack of transparency makes the comparison using their own dataset difficult to evaluate fairly.

・In terms of evaluation metrics, recent makeup transfer research has employed CLIP-based semantic metrics to assess the similarity between two makeup styles at the semantic level. However, this paper lacks such metrics, which reduces the comprehensiveness of its evaluation and makes it less aligned with recent research trends.

**Questions:**

See weakness.

---

### Official Review · Reviewer_XsLR · 2025-10-28

**Soundness:** 3
**Presentation:** 3
**Contribution:** 2
**Rating:** 4
**Confidence:** 4

**Summary:**

This paper presents "MakeupAnyone", a novel framework for the task of makeup transfer, which aims to apply a makeup style from a reference image to a source face while preserving the source's identity. The authors identify two primary challenges in existing methods: 1) the scarcity of high-quality paired data, leading to poor generalization, and 2) the difficulty in perfectly disentangling identity features from makeup style, resulting in facial distortion.
To address these issues, MakeupAnyone proposes a two-pronged solution. First, it introduces a self-supervised data augmentation pipeline that leverages large Vision-Language Models (VLMs) and instruction-guided editing models to generate a vast, high-quality "pseudo-paired" dataset, thus mitigating the data scarcity problem. Second, it designs a Region-Aware Multi-Scale Alignment architecture. This architecture uses two separate encoders—a Makeup Semantic Encoder for global style and a Region Style Encoder for local details—and fuses their features adaptively to guide a diffusion-based generator. A composite loss function, incorporating identity, style, adversarial, and contrastive terms, is used to train the model. The authors demonstrate through extensive experiments that their method achieves state-of-the-art performance in both makeup fidelity and identity preservation.

**Strengths:**

1. The most significant contribution of this paper is its self-supervised data augmentation pipeline. Leveraging modern large-scale generative models (VLMs and instruction-guided editors) to create high-quality, structured pseudo-paired data is a very clever and highly relevant solution to the long-standing data scarcity problem in this domain. The multi-stage quality control (filtering based on identity, aesthetics, and text consistency) further enhances the robustness of this approach.

2. The dual-encoder design (Region-Aware Multi-Scale Alignment) is well-motivated. The idea of explicitly disentangling global semantic style from fine-grained local details addresses a core challenge in makeup transfer. The adaptive fusion module and the use of cross-attention to inject these features into the U-Net provide a principled way to combine these multi-scale representations.

**Weaknesses:**

1 Does the input include both makeup ref image and makeup caption? In my opinion, the makeup caption comes from the VLM. When the makeup captions get wrong, does it lead to a bad makeup result?

2 The inputs of both the region style encoder and the makeup semantic encoder hold the same id with no-makeup image or different ids? If not, where these images come from?

3 It is better for the authors to public their proposed datasets, and give some analysis and statistics about the dataset.  Without publication, it is hard for other researchers to follow the work, which weakens the contribution of the paper.

**Questions:**

See weakness above, Q2&3 are my main concerns.

---

### Official Review · Reviewer_g6uR · 2025-10-31

**Soundness:** 3
**Presentation:** 2
**Contribution:** 2
**Rating:** 4
**Confidence:** 4

**Summary:**

This paper introduces self-supervised makeup transfer framework. To sidestep the dataset requirements, they propose VLM-based pseudo paired synthesis approach, demonstrating high-faithful results.

**Strengths:**

- Paired pseudo makeup dataset: The authors construct pseudo paired makeup images using Qwen2.5-VL’s generated instructions and FLUX.1-Kontext’s editing capabilities. It alleviates the dataset requirement issue in makeup transfer and provides informative supervision signal when training denoising diffusion models.
- Fidelity: The resulting images demonstrated plausible outcomes with high fidelity in terms of structure of source image and makeup effects of reference image.

**Weaknesses:**

- Engineering approach: As mentioned in limitations, this approach heavily rely on existing networks and training strategies. Data augmentation pipeline is reasonable, but the entire framework just combined previous methods and well-known models as engineering manner rather than proposing learning algorithms or efficient tuning strategies, limiting its contribution in this community.
- Semantic loss encoder: The authors mention that their semantic encoder learns a robust visual representation, but there is no details about this encoder structure whether it comes from external components or it is designed itself with training. Furthermore, there are several robust semantic feature extractor like DINO and CLIP encoders. It is wondered that such networks showed lower advantages in experiments or not, and there is any reason to not use above well-known encoders.
- Region style encoder: There is not much detail about how the region style encoder works; how it operates and or what kind of results it produces. Even though regional attention mechanism mainly comes from this path, it has not enough description.
- Lower or on-par identity metric: In quantitative comparison, the system exhibited not much improvements in $CLS$. It is important to discuss why there is no much enhancement in terms of identity despite the various adaptations of existing models.

(Miscellaneous)
- Format and typo: It is imperative to remove Appendix section if there is no supplementary material, which appears in Page 13. Also, some types remain fatal flaws, exacerbating overall quality of the manuscript, e.g., ‘??’ in Sec.3 (Page 3).

**Questions:**

- For semantic loss encoder, provide more detailed description. Which configuration semantic encoder adopt and what advantages it brings compared to existing robust extractor?
- It is recommended to discuss the academic contributions of this paper beyond configured framework and dataset in this community in the perspective of learning representation.
- Is there any criteria or quantifying of in-the-wild condition or real world scenario about makeup style? It seems that the used reference images are not much diverse than the used samples in other papers.
- The authors insist it has signifiant identity preservation, but in terms of $CLS$ metric, the proposed system didn’t show higher performance. It is imperative to discuss how the proposed system brings identity improvements than other approaches.

---

### Official Review · Reviewer_AydU · 2025-11-01

**Soundness:** 2
**Presentation:** 2
**Contribution:** 2
**Rating:** 2
**Confidence:** 4

**Summary:**

This paper introduces an approach (named MakeupAnyone) to deal with the paired data scarcity problem of makeup transfer tasks. This method consists of two stages, i.e., self-supervised data augmentation and region-aware multi-scale alignment. In the first stage, the large vision language models are first used to describe the makeup styles, and the image editing models are then adopted to produce pseudo-makeup images based on the generated captions. While in the second stage, a Makeup Semantic Encoder and a Region Style Encoder are developed to capture and fuse multi-scale global makeup semantics and region-aware style features, respectively.

**Strengths:**

1. The paper is well organized, and the writing of this work is easy to understand. The proposed method is straightforward, and the motivation is well clarified.

2. The quantitative and qualitative comparisons with state-of-the-art methods on three datasets appear to demonstrate the effectiveness of the proposed method.

**Weaknesses:**

1. One of my major concerns lies in the generated pseudo-paired triplets, each includes a source image, a text description, and a target image. Without a real reference image, the proposed method takes the pseudo-makeup images as reference images to generate the makeup condition information that is injected into the denoising U-Net model to control the makeup transfer process. However, in this way, the reference images and source images share similar or even the same facial structural and human identity information, so how to ensure that the target images only capture those information from the source images instead of the reference ones? Such information disentanglement is quite important for makeup transfer, especially for real-world cases where the source and reference images have large pose and expression differences.

2. For the Makeup Semantic Encoder, the authors claim that it can “extract style features that are grounded in textual semantics, thus producing a more abstract and robust style representation.” Why adopting the encoder part of the Denoising U-Net can help to achieve this goal? Could the authors provide a more detailed explanation for this semantic-grounded feature extraction process?

3. Some technical details are missing:

    1) In the Region Style Encoder, the authors perform the region-based augmentations by using the pre-processed makeup area mask. How to generate such a mask? By existing segmentation models or some other techniques?
    2) In the Adaptive Fusion Module, the authors compute spatial and channel attention maps from the concatenated semantic and regional style features. Could the authors provide the calculation operations for these two attention maps?
    3) For the Adversarial Loss in Eq. (3), there is only the objective function for the generated images. Does the discriminator require updating using the real makeup images?

4. For the experimental results, the authors claim that their collected dataset contains “makeup images cover a wide range of styles from daily light makeup to heavy artistic makeup” and “featuring rich diversity in pose and expression”. But the qualitative comparisons only present the results with simple makeup styles, and small pose and expression variants between source and reference images. Could the authors provide more visualization results of complex makeup styles, such as those in LADN and CPM datasets? And with large pose and expression variants?

5. I am wondering whether the proposed method can handle some real-world cases, for example, the source and reference images contain occlusions, skin color differences,  transferring only local makeup style (i.e., eyes, lips, faces), or combining several local styles from different reference images.

**Questions:**

Please refer to Weaknesses.

---

### Meta-Review · Area_Chair_1QWT · 2025-12-09

**Summary:**

This paper introduces a self-supervised framework for makeup transfer. All reviewers recommended rejection, with scores of 2, 4, 4, and 2. The reviewers are mainly concerned about the novelty and the technical sound of the paper. Specifically, Reviewer AydU and Reviewer XsLR ask for details regarding the Makeup Semantic and Region Style Encoders, etc. Moreover, Reviewer MY1A questions the "self-supervised" label, arguing that the method still relies on supervised training via synthetic data. Furthermore, the authors did not provide a rebuttal to address all the concerns. Thus, the AC recommends rejecting this paper.

**Reviewer Concerns:**

As the authors did not provide any rebuttal, the concerns raised during the review process are still outstanding.

**Reviewer Scores:**

As no rebuttal was submitted, all the reviewers are unlikely to raise their scores.

---

### Decision · Program_Chairs · 2026-01-26

Reject